# Rethinking Probabilistic Circuit Parameter Learning

Anji Liu[1]                    Guy Van den Broeck[1]

[1]Department of Computer Science, UCLA, Los Angeles, California, USA

## Abstract

Probabilistic Circuits (PCs) offer a computationally scalable framework for generative modeling, supporting exact and efficient inference of a wide range of probabilistic queries. While recent advances have significantly improved the expressiveness and scalability of PCs, effectively training their parameters remains a challenge. In particular, a widely used optimization method, full-batch Expectation-Maximization (EM), requires processing the entire dataset before performing a single update, making it ineffective for large datasets. While empirical extensions to the mini-batch setting have been proposed, it remains unclear what objective these algorithms are optimizing, making it difficult to assess their theoretical soundness. This paper bridges the gap by establishing a novel connection between the general EM objective and the standard full-batch EM algorithm. Building on this, we derive a theoretically grounded generalization to the mini-batch setting and demonstrate its effectiveness through preliminary empirical results.

## 1 INTRODUCTION

Probabilistic Circuits (PCs) are a class of generative models that represent probability distributions by recursively composing simpler distributions through sum (mixture) and product (factorization) operations [Choi et al., 2020]. The key idea behind PCs is to examine how tractable probabilistic models, such as Hidden Markov Models [Rabiner and Juang, 1986], perform inference (e.g., computing marginal probabilities). PCs distill the structure of these models' computation graphs into a compact and general framework, which leads to a unified, computation-oriented perspective on tractable probabilistic modeling.

While significant progress has been made in improving the expressiveness of PCs through architectural innovations [Loconte et al., 2025, Liu and Van den Broeck, 2021] and system-level advancements [Liu et al., 2024, Peharz et al., 2020], there is still no clear consensus on how to effectively learn their parameters. Full-batch Expectation-Maximization (EM) and its empirical variants remain widely used approaches [Zhang et al., 2025, Liu et al., 2023]. However, full-batch EM requires aggregating information across the entire dataset before each parameter update, making it hard to scale to large datasets or streaming settings.

Based on Kunstner et al. [2021], which studies the full-batch EM algorithm for all latent variable models, we discover that the full-batch EM update of PCs corresponds to optimizing a 1st order Taylor approximation, regularized by a Kullback–Leibler (KL) divergence that penalizes deviation from the current distribution. This yields a novel view of the full-batch EM update for PCs, which has appeared in various forms across different contexts [Peharz, 2015, Choi et al., 2021, Poon and Domingos, 2011].

This perspective naturally suggests a theoretically grounded mini-batch extension: by increasing the weight on the KL term, we can compensate for the reduced information available in a mini-batch compared to the full dataset. The resulting update rule admits a closed-form expression, making it efficient and easy to implement. Preliminary empirical evaluations further demonstrate the empirical superiority compared to existing EM- and gradient-based algorithms.

## 2 BACKGROUND

### 2.1 DISTRIBUTIONS AS CIRCUITS

Probabilistic Circuits (PCs) represent probability distributions with deep and structured computation graphs that consist of sum and product operations [Choi et al., 2020]. They are an extension of existing tractable probabilistic models, which are designed to support efficient and exact probabilistic inference over complex queries, such as Sum Product

Networks [Poon and Domingos, 2011], cutset networks [Rahman et al., 2014], and Hidden Markov Models [Rabiner and Juang, 1986]. PCs inherit the tractability of these models while introducing a scalable computational framework to encode richer and more complex data distributions. The syntax and semantics of PCs are as follows:

**Definition 1** (Probabilistic Circuit). A PC $p$ over variables $\mathbf{X}$ is a directed acyclic computation graph with one single root node $n_{\mathrm{r}}$. Every input node (those without incoming edges) in $p$ defines an univariate distribution over variable $X \in \mathbf{X}$. Every inner node (those with incoming edges) is either a *product* or a *sum* node, where each product node encodes a factorized distribution over its child distributions and each sum node represents a weighted mixture of its child distributions. Formally, the distribution $p_n$ encoded by a node $n$ can be represented recursively as

$$p_n(\boldsymbol{x}) := \begin{cases} f_n(\boldsymbol{x}) & n \text{ is an input node,} \\ \prod_{c \in \mathsf{ch}(n)} p_c(\boldsymbol{x}) & n \text{ is a product node,} \\ \sum_{c \in \mathsf{ch}(n)} \theta_{n,c} \cdot p_c(\boldsymbol{x}) & n \text{ is a sum node,} \end{cases} \quad (1)$$

where $f_n$ is an univariate primitive distribution defined over $X \in \mathbf{X}$ (e.g., Gaussian, Categorical), $\mathsf{ch}(n)$ denotes the set of child nodes of $n$, and $\theta_{n,c} \geq 0$ is the parameter corresponds to the edge $(n, c)$ in the PC. Define the log-parameter of $(n, c)$ as $\phi_{n,c} := \log \theta_{n,c}$, which will be used interchangeably with $\theta_{n,c}$. We further denote $\phi := \{\phi_{n,c}\}_{(n,c)}$ as the set of all sum node parameters in the PC. Without loss of generality, we assume that every path from the root node to an input node alternates between sum and product nodes.

In order to ensure exact and efficient computation of various probabilistic queries, including marginalization and computing moments, we need to impose structural constraints on the circuit. Specifically, smoothness and decomposability [Peharz et al., 2015] are a set of sufficient conditions that ensure tractable computation of marginal and conditional probabilities. Intuitively, this tractability arises because smooth and decomposable circuits represent multilinear functions, which are known to support efficient marginalization [Broadrick et al., 2024]. We provide further details in Appendix A.

PCs can be viewed as latent variable models with discrete latent spaces [Peharz et al., 2016]. Each sum node can be interpreted as introducing a discrete latent variable $Z$ that selects among its child distributions. Specifically, assigning $Z = i$ corresponds to choosing the $i$-th child of the sum node. By aggregating all such latent variables, the PC can be seen as defining a hierarchical latent variable model.

## 2.2 EXPECTATION-MAXIMIZATION

Expectation-Maximization (EM) is a well-known algorithm to maximize the log-likelihood given data $\boldsymbol{x}$ of a distribution defined by a latent variable model. Specifically, the

distribution $p_\phi(\mathbf{X})$ ($\phi$ are the parameters) is defined as $\sum_{\boldsymbol{z}} p_\phi(\mathbf{X}, \boldsymbol{z})$ over latents $\mathbf{Z}$. Our goal is to maximize

$$\mathtt{LL}(\boldsymbol{\phi}) := \log p_\phi(\boldsymbol{x}) = \log\Big(\sum_{\boldsymbol{z}} p_\phi(\boldsymbol{x}, \boldsymbol{z})\Big). \quad (2)$$

EM is an effective way to maximize the above objective when $p_\phi(\mathbf{X}, \mathbf{Z})$ permits much simpler (or even closed-form) maximum likelihood estimation. It optimizes Equation (2) by maximizing the following surrogate objective:

$$Q_\phi(\phi') := \sum_{\boldsymbol{z}} p_\phi(\boldsymbol{z}|\boldsymbol{x}) \cdot \log p_{\phi'}(\boldsymbol{x}, \boldsymbol{z}). \quad (3)$$

Given the current set of parameters $\phi$, EM updates the parameters by solving for $\phi'$ that maximizes $Q_\phi(\phi')$, which is guaranteed to be a lower bound of $\mathtt{LL}(\phi')$ since

$$Q_\phi(\phi') = \mathtt{LL}(\phi') + \sum_{\boldsymbol{z}} p_\phi(\boldsymbol{z}|\boldsymbol{x}) \cdot \log p_{\phi'}(\boldsymbol{z}|\boldsymbol{x}) \leq \mathtt{LL}(\phi').$$

## 3 EM FOR PROBABILISTIC CIRCUITS

While variants of the EM algorithm have been proposed for training PCs in various contexts [Poon and Domingos, 2011, Peharz, 2015], their connection to the general EM objective $Q_\phi(\phi')$ (cf. Eq. (3)) remains implicit. The lack of a unified formulation makes it difficult to fully understand the existing optimization procedures or to extend them to new settings, such as training with mini-batches of data, which is critical for scaling the optimizer to large datasets.

Specifically, there are multiple ways to define mini-batch EM algorithms that all reduce to the same full-batch EM algorithm in the limit. However, it is often unclear what objective these variants are optimizing in the mini-batch case, which complicates the design of new learning algorithms.

In this section, we bridge this gap by deriving EM for PCs explicitly from the general objective. In Section 3.1, we begin with a derivation for the full-batch case, showing how existing formulations can be recovered and interpreted from this viewpoint. We then extend the derivation to the mini-batch setting in Section 3.2, leading to a principled and theoretically-grounded mini-batch EM algorithm for PCs.

### 3.1 REVISITING FULL-BATCH EM

Recall from Definition 1 that we define the log-parameter that corresponds to the edge $(n, c)$ as $\phi_{n,c} := \log \theta_{n,c}$, and the set of all parameters of a PC as $\phi := \{\phi_{n,c}\}_{n,c}$.[1] Since $\phi$ does not necessarily define a normalized PC, we distinguish between the unnormalized and normalized forms of

---

[1]We assume for simplicity that distributions of input nodes have no parameters (e.g., indicator distributions). Our analysis can be easily extended to exponential family input distributions.

the model: let $\tilde{p}_\phi(\boldsymbol{x})$ denote the unnormalized output of the circuit computed via the feedforward pass defined by Equation (1), and define the normalized distribution as

$$p_\phi(\boldsymbol{x}) := \tilde{p}_\phi(\boldsymbol{x})/Z(\phi),$$

where $Z(\phi)$ is the normalizing constant. We extend the single-sample EM objective in Equation (3) to the following, which is defined on a dataset $\mathcal{D}$:

$$Q_\phi^{\mathcal{D}}(\phi') := \frac{1}{|\mathcal{D}|} \sum_{\boldsymbol{x}\in\mathcal{D}} \sum_{\boldsymbol{z}} p_\phi(\boldsymbol{z}|\boldsymbol{x}) \cdot \log p_{\phi'}(\boldsymbol{x},\boldsymbol{z}).$$

Our analysis is rooted in the following result.

**Proposition 1.** *Given a PC $p_\phi$ with log-parameters $\phi$ (cf. Def. 1) and a dataset $\mathcal{D}$, the objective $Q_\phi^{\mathcal{D}}(\phi')$ equals the following up to a constant term irrelevant to $\phi'$:*

$$\frac{1}{|\mathcal{D}|}\sum_{\boldsymbol{x}\in\mathcal{D}}\underbrace{\log p_\phi(\boldsymbol{x}) + \left\langle \frac{\partial \log p_\phi(\boldsymbol{x})}{\partial\phi}, \phi'-\phi \right\rangle}_{\texttt{LinLL}_\phi^{\boldsymbol{x}}(\phi')} - \texttt{KL}_\phi(\phi'),$$

*where $\texttt{KL}_\phi(\phi') := \mathrm{D}_{\mathrm{KL}}\left(p_\phi(\mathbf{X},\mathbf{Z}) \,\|\, p_{\phi'}(\mathbf{X},\mathbf{Z})\right)$ is the KL divergence between $p_\phi$ and $p_{\phi'}$.*

The proof follows Kunstner et al. [2021] and is provided in Appendix B.1. Proposition 1 reveals that the EM update can be interpreted as maximizing a *regularized first-order approximation* of the log-likelihood. Specifically, the term $\texttt{LinLL}_\phi^{\boldsymbol{x}}(\phi')$ corresponds to the linearization of $\log p_{\phi'}(\boldsymbol{x})$ around the current parameters $\phi$, capturing the local sensitivity of the log-likelihood to parameter changes. The KL term, $\texttt{KL}_\phi(\phi')$, acts as a regularizer that penalizes large deviations in the joint distribution over $\mathbf{X}$ and $\mathbf{Z}$.

According to Proposition 1, solving for the updated parameters $\phi'$ requires computing two key quantities: (i) the gradient of the log-likelihood $\partial \log p_\phi(\boldsymbol{x})/\partial\phi$, and the KL divergence $\texttt{KL}_\phi(\phi')$. To express these terms in closed form, we introduce the concept of top-down probabilities, which is first defined by Dang et al. [2022].

**Definition 2** (TD-prob). Given a PC $p$ parameterized by $\phi$, we define the top-down probability (TD-prob) $\texttt{TD}(n)$ of a node $n$ recursively from the root node to input nodes:

$$\texttt{TD}(n) := \begin{cases} 1 & n \text{ is the root node,} \\ \sum_{m\in\texttt{pa}(n)} \texttt{TD}(m) & n \text{ is a sum node,} \\ \sum_{m\in\texttt{pa}(n)} \theta_{m,n}\cdot\texttt{TD}(m) & n \text{ is a product node,} \end{cases}$$

where $\theta_{m,n} := \exp(\phi_{m,n})$ and $\texttt{pa}(n)$ is the set of parent nodes of $n$. Define the TD-prob of $\phi_{m,n}$ as $\texttt{TD}(\phi_{m,n}) := \theta_{m,n}\cdot\texttt{TD}(m)$, and denote by $\texttt{TD}(\phi)$ the vector containing the TD-probs of all edge parameters in the circuit.

Intuitively, the TD-prob of a parameter quantifies how much influence it has on the overall output of the PC, and in

particular, on the normalizing constant $Z(\phi)$.[2] We continue to express the two key terms in Proposition 1 in closed form.

**Lemma 1.** *Assume the distributions defined by all nodes in a PC are normalized. We have the following ($\forall\boldsymbol{x}$):*

*(i)* $\partial \log p_\phi(\boldsymbol{x})/\partial\phi = \partial \log \hat{p}_\phi(\boldsymbol{x})/\partial\phi - \texttt{TD}(\phi),$

*(ii)* $\texttt{KL}_\phi(\phi') = -\langle\texttt{TD}(\phi),\phi'\rangle + C,$

*where $C$ is a constant term independent of $\phi'$.*

The assumption that the PC is normalized is mild and practical. In Appendix C, we introduce a simple and efficient algorithm that adjusts the PC parameters to ensure normalization without affecting the structure of the circuit. We can now substitute the closed-form expressions for the gradient and the KL divergence into the general EM objective $Q_\phi^{\mathcal{D}}(\phi')$, which converts the problem into[3]

$$\left\langle \frac{1}{|\mathcal{D}|}\sum_{\boldsymbol{x}\in\mathcal{D}} \frac{\partial\log\tilde{p}_\phi(\boldsymbol{x})}{\partial\phi} - \cancel{\texttt{TD}(\phi)}, \phi'\right\rangle + \cancel{\langle\texttt{TD}(\phi),\phi'\rangle}.$$

If we additionally require each node in the PC to define a normalized distribution, we impose the constraint $\sum_{c\in\texttt{ch}(n)}\exp(\phi'_{n,c}) = 1$ for all sum nodes $n$. Incorporating these constraints into the EM objective results in a constrained maximization problem that has the following solution for every edge $(n,c)$ (see Appx. B.2 for the derivation):

$$\phi'_{n,c} = \log\theta'_{n,c}, \quad \theta'_{n,c} = \texttt{F}_\phi^{\mathcal{D}}(n,c)/Z, \qquad (4)$$

where we define $\texttt{F}_\phi^{\mathcal{D}}(n,c) := \frac{1}{|\mathcal{D}|}\sum_{\boldsymbol{x}\in\mathcal{D}}\frac{\partial\log\tilde{p}_\phi(\boldsymbol{x})}{\partial\phi_{n,c}}$[4], and $Z = \sum_{c'\in\texttt{ch}(n)}\texttt{F}_\phi^{\mathcal{D}}(n,c')$ ensures that $n$ is normalized.

While this full-batch EM algorithm in Equation (4) has been derived in prior work [Choi et al., 2021, Peharz, 2015], we recover it here through Proposition 1. This paves the way for a principled mini-batch EM algorithm by generalizing the objective $Q_\phi^{\mathcal{D}}(\phi')$, as shown in the next section.

### 3.2 EXTENSION TO THE MINI-BATCH CASE

When the dataset is large, full-batch EM becomes inefficient and impractical as it requires scanning the entire dataset before making any parameter updates. In such cases, we instead wish to update the parameters after processing only a small subset of data points, which is commonly referred to as a mini-batch. Given a mini-batch of samples $\mathcal{D}$, Peharz et al. [2020] proposes to update the parameters according to a small step size $\alpha\in(0,1)$:

$$\theta'_{n,c} = (1-\alpha)\cdot\theta_{n,c} + \alpha\cdot\texttt{F}_\phi^{\mathcal{D}}(n,c)/Z, \qquad (5)$$

---

[2]This can be observed from the fact that $Z(\phi)$ can be computed via the same feedforward pass (Eq. (1)) except that we set the output of input nodes to 1.

[3]We drop all terms that are independent of $\phi'$.

[4]This corresponds to the PC flows defined by Choi et al. [2021].

where we borrow notation from Equation (4). However, it remains unclear whether this update rule is grounded in a principled EM objective. In the following, we show that from the full-batch EM derivation in the previous section, we can derive a mini-batch update rule that closely resembles the above, but with a crucial difference.

Proposition 1 expresses the EM objective as the sum of two terms: a linear approximation of the log-likelihood and a regularization term that penalizes deviation from the current model via KL divergence. When using only a mini-batch of samples, the log-likelihood may overlook parts of the data distribution not covered by the sampled subset.

To account for this, we can put a weighting factor $\gamma > 1$ on the KL divergence (i.e., $\text{KL}_\phi(\phi')$ becomes $\gamma \cdot \text{KL}_\phi(\phi')$).[5] Plugging in Lemma 1 and dropping terms independent to $\phi'$, the adjusted objective (i.e., $Q_\phi^{\mathcal{D}}(\phi')$ with the additional weighting $\gamma$) can be expressed as

$$\langle \text{F}_\phi^{\mathcal{D}}, \phi' \rangle + (\gamma - 1) \cdot \langle \text{TD}(\phi), \phi' \rangle,$$

where $\text{F}_\phi^{\mathcal{D}}$ collects all entries $\text{F}_\phi^{\mathcal{D}}(n, c)$ into a single vector, with each $\text{F}_\phi^{\mathcal{D}}(n, c)$ representing the aggregated gradient w.r.t. $\phi_{n,c}$. With the constraints that ensure each PC node defines a normalized distribution, the solution is

$$\theta'_{n,c} = \left( \text{TD}_\phi(n) \cdot \theta_{n,c} + \alpha \cdot F_\phi^{\mathcal{D}}(n, c) \right) / Z, \quad (6)$$

where $\alpha := 1/(\gamma - 1)$ is the learning rate, $\text{TD}_\phi(n)$ is the TD-prob of node $n$ (Def. 2), and $Z$ is a normalizing constant. The derivation is deferred to Appendix B.2. In practice, compared to the full-batch EM update (Eq. (4)), the only additional computation required is $\text{TD}_\phi(n)$, which can be efficiently computed once in a data-independent top-down pass over the PC using Definition 2.

To build intuition for the update rule, we consider the case where $\mathcal{D}$ contains only a single sample $\boldsymbol{x}$. In this setting, the update direction $\text{F}_\phi^{\boldsymbol{x}}(n, c)$ can be decomposed using the chain rule of derivatives:

$$\text{F}_\phi^{\boldsymbol{x}}(n, c) = \frac{\partial \log \tilde{p}_\phi(\boldsymbol{x})}{\partial \phi_{n,c}} = \underbrace{\frac{\partial \log \tilde{p}_\phi(\boldsymbol{x})}{\partial \log \tilde{p}_\phi^n(\boldsymbol{x})}}_{\text{F}_\phi^{\boldsymbol{x}}(n)} \cdot \underbrace{\frac{\partial \log \tilde{p}_\phi^n(\boldsymbol{x})}{\partial \phi_{n,c}}}_{\hat{\text{F}}_\phi^{\boldsymbol{x}}(n,c)},$$

where we define $\log \tilde{p}_\phi^n(\boldsymbol{x})$ as the (unnormalized) log-likelihood of node $n$. A key observation is that the second term $\hat{\text{F}}_\phi^{\boldsymbol{x}}(n, c)$ is normalized w.r.t. all children of sum node $n$: $\sum_{c \in \text{ch}(n)} \hat{\text{F}}_\phi^{\boldsymbol{x}}(n, c) = 1$ (see Appx. B.3 for the proof). Intuitively, we now break down $\text{F}_\phi^{\boldsymbol{x}}(n, c)$ into the *importance of node* $n$ to the overall output (i.e., $\text{F}_\phi^{\boldsymbol{x}}(n)$) and the *relative contribution of child* $c$ to $n$ (i.e., $\hat{\text{F}}_\phi^{\boldsymbol{x}}(n, c)$). With this decomposition, we can simplify Equation (6) as

$$\theta'_{n,c} = \left( \theta_{n,c} + \alpha \cdot \text{rel}_\phi^{\boldsymbol{x}}(n) \cdot \hat{\text{F}}_\phi^{\boldsymbol{x}}(n, c) \right) / Z,$$

---

[5]Note that this is equivalent to $Q_\phi^{\mathcal{D}}(\phi') - (\gamma - 1) \cdot \text{KL}_\phi(\phi')$ according to Proposition 1.

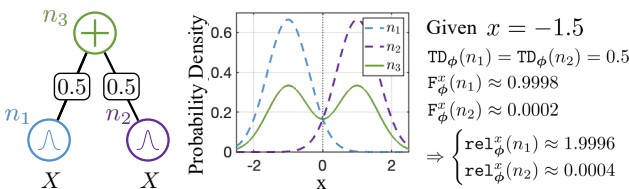

Figure 1: The proposed algorithm implicitly applies an adaptive learning rate to each node. For the PC shown on the left, given a sample $x = -1.5$, the algorithm uses a large learning rate to update $n_1$ while keeping $n_2$ almost unchanged.

where $\text{rel}_\phi^{\boldsymbol{x}}(n) := \text{F}_\phi^{\boldsymbol{x}}(n)/\text{TD}_\phi(n)$ can be viewed as the relative importance of $n$ to the PC output given input $\boldsymbol{x}$. The term $\alpha \cdot \text{rel}_\phi^{\boldsymbol{x}}(n)$ then acts as an adaptive learning rate for updating the child parameters of node $n$, scaling the update magnitude according to how influential $n$ is for the input. In comparison, the mini-batch algorithm in Equation (5) uses a fixed learning rate for all parameters.

This difference is reflected in the example shown in Figure 1. Given the PC on the left, which represents a mixture of two Gaussians (middle). Suppose we draw one sample $x = -1.5$. This sample does not reflect the full distribution and only activates the left mode. Our algorithm accounts for this by assigning a small effective learning rate to node $n_2$ ($\text{rel}_\phi^x(n_2) \approx 0.0004$) as it is "not responsible" for explaining this particular input and focuses the update on $n_1$ ($\text{rel}_\phi^x(n_1) \approx 1.9996$). In contrast, Equation (5) applies a uniform learning rate across all parameters, leading it to also update $n_2$ unnecessarily to fit the current sample.

## 4 CONNECTIONS WITH GRADIENT-BASED OPTIMIZERS

Gradient-based optimization methods, such as stochastic gradient descent (SGD), can be interpreted through a lens similar to the EM formulation. Recall from Proposition 1 that the EM objective comprises a linear approximation of the log-likelihood, along with a KL divergence regularizer that penalizes deviations from the current model. In contrast, standard gradient-based updates can be viewed as maximizing the same linear approximation of the log-likelihood, but with an *L2 regularization* penalty on parameter updates instead of a KL divergence:

$$\frac{1}{|\mathcal{D}|} \sum_{\boldsymbol{x} \in \mathcal{D}} \log p_\phi(\boldsymbol{x}) + \left\langle \frac{\partial \log p_\phi(\boldsymbol{x})}{\partial \phi}, \phi' - \phi \right\rangle - \gamma \|\phi' - \phi\|_2^2.$$

Solving for $\phi'$ gives $\phi' = \phi + \alpha \cdot \partial \log p_\phi(\boldsymbol{x})/\partial \phi$, a standard gradient ascent step with $\alpha = 1/(2\gamma)$.

We empirically compare our mini-batch EM algorithm against existing EM- and gradient-based optimizers on training a PC to fit the ImageNet32 dataset [Deng et al., 2009], where the proposed algorithm converges faster and also achieves higher likelihoods after convergence (Appx. D).

## Acknowledgements

This work was funded in part by the DARPA ANSR, CODORD, and SAFRON programs under awards FA8750-23-2-0004, HR00112590089, and HR00112530141, NSF grant IIS1943641, and gifts from Adobe Research, Cisco Research, and Amazon. Approved for public release; distribution is unlimited.

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

# Rethinking Probabilistic Circuit Parameter Learning
## (Supplementary Material)

**Anji Liu**[1]          **Guy Van den Broeck**[1]

[1]Department of Computer Science, UCLA, Los Angeles, California, USA

## A    STRUCTURAL PROPERTIES OF PCS

We provide formal definitions of smoothness and decomposability. Please refer to Choi et al. [2020] for a comprehensive overview.

**Definition 3** (Smoothness and Decomposability). Define the scope $\text{Var}(n)$ of a PC node $n$ as the set of all variables defined by its descendant input nodes. A PC $p$ is smooth if for every sum node $n$, its children share the same scope: $\forall c_1, c_2 \in \text{ch}(n)$, $\text{Var}(c_1) = \text{Var}(c_2)$. $p$ is decomposable if for every product node $n$, its children have disjoint scopes: $\forall c_1, c_2 \in \text{ch}(n)$ $(c_1 \neq c_2)$, $\text{Var}(c_1) \cap \text{Var}(c_2) = \varnothing$.

## B    PROOFS

This section provides proof of the theoretical results stated in the main paper.

### B.1    INTERPRETING THE EM ALGORITHM OF PCS

This section provides the proof of Proposition 1, which interprets the full-batch EM algorithm of PCs in a new context.

*Proof of Proposition 1.* We begin by formalizing the latent-variable-model view of PCs. Given a PC $p_{\boldsymbol{\phi}}(\mathbf{X})$ parameterized by $\boldsymbol{\phi}$, we define a set of latent variables $\mathbf{Z}$ such that $p_{\boldsymbol{\phi}}(\mathbf{X}) = \sum_{\boldsymbol{z}} p_{\boldsymbol{\phi}}(\mathbf{X}, \mathbf{Z} = \boldsymbol{z})$. Specifically, we associate a latent variable $Z_n$ with each sum node $n$ in the PC. We use $Z_n = i$ ($i \in \{1, \ldots, |\text{ch}(n)|\}$) to denote that we "select" the $i$-th child node by zeroing out all the probabilities coming from all other child nodes:

$$p_n(\boldsymbol{x}, Z_n = i, \boldsymbol{z}_{\backslash n}) = \sum_{c \in \text{ch}(n)} \exp(\phi_{n,c}) \cdot p_c(\boldsymbol{x}, Z_n = i, \boldsymbol{z}_{\backslash n}) \cdot \mathbb{1}[c = c_i],$$

where we define $c_i$ as the $i$-th child node of $n$, and $\mathbf{Z}_{\backslash n} := \mathbf{Z} \backslash Z_n$.

We further show that $p_{\boldsymbol{\phi}}(\mathbf{X}, \mathbf{Z})$ is an exponential family distribution. To see this, it suffices to construct a set of $|\boldsymbol{\phi}|$ sufficient statistics $S(\boldsymbol{x}, \boldsymbol{z})$ such that for every $\boldsymbol{x}$ and $\boldsymbol{z}$, the likelihood can be expressed as:

$$p_{\boldsymbol{\phi}}(\boldsymbol{x}, \boldsymbol{z}) = \exp\left(\langle S(\boldsymbol{x}, \boldsymbol{z}), \boldsymbol{\phi} \rangle - A(\boldsymbol{\phi})\right),$$

where $A(\boldsymbol{\phi}) = \log \sum_{\boldsymbol{x}, \boldsymbol{z}} \langle S(\boldsymbol{x}, \boldsymbol{z}), \boldsymbol{\phi} \rangle$ is the log partition function that normalizes the distribution. Note that $A(\boldsymbol{\phi})$ is convex by definition.

To construct $S(\boldsymbol{x}, \boldsymbol{z})$, we first define the *support* $\text{supp}(n)$ of every node recursively as follows:

$$\text{supp}(n) := \begin{cases} \{(\boldsymbol{x}, \boldsymbol{z}) : p_n(\boldsymbol{x}) > 0\} & n \text{ is an input node}, \\ \bigcap_{c \in \text{ch}(n)} \text{supp}(c) & n \text{ is a product node}, \\ \bigcup_{c \in \text{ch}(n)} \left(\{(\boldsymbol{x}, \boldsymbol{z}) : z_n = c\} \cap \text{supp}(c)\right) & n \text{ is a sum node}, \end{cases}$$

where $z_n = c$ means $z_n = i$ if $c$ is the $i$-th child of $n$.

The sufficient statistics $S(\boldsymbol{x}, \boldsymbol{z})$ can be defined using the support. Specifically, the sufficient statistics corresponding to the parameter $\phi_{n,c}$, denoted $S_{\phi_{n,c}}(\boldsymbol{x}, \boldsymbol{z})$, can be represented as:

$$S_{\phi_{n,c}}(\boldsymbol{x}, \boldsymbol{z}) = \mathbb{1}\Big[(\boldsymbol{x}, \boldsymbol{z}) \in \mathrm{supp}(c) \text{ and } z_n = c\Big],$$

where $\mathbb{1}[\cdot]$ is the indicator function.

Before proceeding, we define the Bregman divergence induced by a convex function $h$ as:

$$D_h(\boldsymbol{y}, \boldsymbol{x}) := h(\boldsymbol{y}) - h(\boldsymbol{x}) - \left\langle \frac{\partial h(\boldsymbol{x})}{\partial \boldsymbol{x}}, \boldsymbol{y} - \boldsymbol{x} \right\rangle.$$

The following part partially follows Kunstner et al. [2021]. We plug in the exponential family distribution form of the PC into the definition of $Q_\phi(\phi')$:

$$
\begin{aligned}
Q_\phi(\phi') &= \frac{1}{|\mathcal{D}|} \sum_{\boldsymbol{x} \in \mathcal{D}} \sum_{\boldsymbol{z}} p_\phi(\boldsymbol{z}|\boldsymbol{x}) \log p_{\phi'}(\boldsymbol{x}, \boldsymbol{z}), \\
&= \frac{1}{|\mathcal{D}|} \sum_{\boldsymbol{x} \in \mathcal{D}} \sum_{\boldsymbol{z}} p_\phi(\boldsymbol{z}|\boldsymbol{x}) \left[ \langle S(\boldsymbol{x}, \boldsymbol{z}), \phi' \rangle - A(\phi') \right], && \triangleright \text{Definition of } p_{\phi'}(\boldsymbol{x}, \boldsymbol{z}) \\
&= \frac{1}{|\mathcal{D}|} \sum_{\boldsymbol{x} \in \mathcal{D}} \left\langle \mathbb{E}_{p_\phi(\boldsymbol{z}|\boldsymbol{x})} \left[ S(\boldsymbol{x}, \boldsymbol{z}) \right], \phi' \right\rangle - A(\phi'). && \triangleright \text{Linearity of expectation}
\end{aligned}
$$

We then subtract both sides by $Q_\phi(\phi)$, which is irrelevant to $\phi'$:

$$
\begin{aligned}
Q_\phi(\phi') - Q_\phi(\phi) &= \frac{1}{|\mathcal{D}|} \sum_{\boldsymbol{x} \in \mathcal{D}} \left\langle \mathbb{E}_{\boldsymbol{z} \sim p_\phi(\cdot|\boldsymbol{x})} \left[ S(\boldsymbol{x}, \boldsymbol{z}) \right], \phi' - \phi \right\rangle - A(\phi') + A(\phi), \\
&= \frac{1}{|\mathcal{D}|} \sum_{\boldsymbol{x} \in \mathcal{D}} \left\langle \mathbb{E}_{\boldsymbol{z} \sim p_\phi(\cdot|\boldsymbol{x})} \left[ S(\boldsymbol{x}, \boldsymbol{z}) \right] - \frac{\partial A(\phi)}{\partial \phi}, \phi' - \phi \right\rangle - \underbrace{\left( A(\phi') - A(\phi) - \left\langle \frac{\partial A(\phi)}{\partial \phi}, \phi' - \phi \right\rangle \right)}_{D_A(\phi', \phi)}, \\
&= \frac{1}{|\mathcal{D}|} \sum_{\boldsymbol{x} \in \mathcal{D}} \left\langle \mathbb{E}_{\boldsymbol{z} \sim p_\phi(\cdot|\boldsymbol{x})} \left[ S(\boldsymbol{x}, \boldsymbol{z}) \right] - \frac{\partial A(\phi)}{\partial \phi}, \phi' - \phi \right\rangle - D_A(\phi', \phi). && (7)
\end{aligned}
$$

We continue by simplifying the first term in the above expression. To do this, consider the gradient of $\mathrm{LL}(\phi)$ w.r.t. $\phi$:

$$
\begin{aligned}
\frac{\partial \mathrm{LL}(\phi)}{\partial \phi} &= \frac{1}{|\mathcal{D}|} \sum_{\boldsymbol{x} \in \mathcal{D}} \frac{\partial \log p_\phi(\boldsymbol{x})}{\partial \phi}, \\
&= \frac{1}{|\mathcal{D}|} \sum_{\boldsymbol{x} \in \mathcal{D}} \frac{\partial \log \left( \sum_{\boldsymbol{z}} \exp \left( \langle S(\boldsymbol{x}, \boldsymbol{z}), \phi \rangle \right) \right)}{\partial \phi} - \frac{\partial A(\phi)}{\partial \phi}, \\
&= \frac{1}{|\mathcal{D}|} \sum_{\boldsymbol{x} \in \mathcal{D}} \sum_{\boldsymbol{z}} \frac{\exp \left( \langle S(\boldsymbol{x}, \boldsymbol{z}), \phi \rangle \right) \cdot S(\boldsymbol{x}, \boldsymbol{z})}{\sum_{\boldsymbol{z}'} \exp \left( \langle S(\boldsymbol{x}, \boldsymbol{z}'), \phi \rangle \right)} - \frac{\partial A(\phi)}{\partial \phi}, \\
&= \mathbb{E}_{\boldsymbol{x} \sim \mathcal{D}, \boldsymbol{z} \sim p_\phi(\cdot|\boldsymbol{x})} \left[ S(\boldsymbol{x}, \boldsymbol{z}) \right] - \frac{\partial A(\phi)}{\partial \phi}.
\end{aligned}
$$

Plug in Equation (7), we have

$$Q_\phi(\phi') - Q_\phi(\phi) = \left\langle \frac{\partial \mathrm{LL}(\phi)}{\partial \phi}, \phi' - \phi \right\rangle - D_A(\phi', \phi). \qquad (8)$$

We proceed to demonstrate that $D_A(\phi', \phi) = D_{\mathrm{KL}}(p_\phi(\mathbf{X}, \mathbf{Z}) \,\|\, p_{\phi'}(\mathbf{X}, \mathbf{Z}))$, where $D_{\mathrm{KL}}(p \,\|\, q)$ is the KL divergence between distributions $p$ and $q$:

$$D_{\mathrm{KL}}(p_\phi(\mathbf{X}, \mathbf{Z}) \,\|\, p_{\phi'}(\boldsymbol{x}, \boldsymbol{z})) = \mathbb{E}_{\boldsymbol{x}, \boldsymbol{z} \sim p_\phi} \left[ \log \frac{p_\phi(\boldsymbol{x}, \boldsymbol{z})}{p_{\phi'}(\boldsymbol{x}, \boldsymbol{z})} \right],$$

$$
\begin{aligned}
&= \mathbb{E}_{\boldsymbol{x},\boldsymbol{z}\sim p_{\boldsymbol{\phi}}}\left[\langle S(\boldsymbol{x},\boldsymbol{z}),\boldsymbol{\phi}-\boldsymbol{\phi}'\rangle\right] + A(\boldsymbol{\phi}') - A(\boldsymbol{\phi}), \\
&= \langle \mathbb{E}_{\boldsymbol{x},\boldsymbol{z}\sim p_{\boldsymbol{\phi}}}\left[S(\boldsymbol{x},\boldsymbol{z})\right],\boldsymbol{\phi}-\boldsymbol{\phi}'\rangle + A(\boldsymbol{\phi}') - A(\boldsymbol{\phi}), \\
&= \left\langle \frac{\partial A(\boldsymbol{\phi})}{\partial \boldsymbol{\phi}},\boldsymbol{\phi}-\boldsymbol{\phi}'\right\rangle + A(\boldsymbol{\phi}') - A(\boldsymbol{\phi}), \qquad\qquad \triangleright \text{ Since } \frac{\partial A(\boldsymbol{\phi})}{\partial \boldsymbol{\phi}} = \mathbb{E}_{\boldsymbol{x},\boldsymbol{z}\sim p_{\boldsymbol{\phi}}}\left[S(\boldsymbol{x},\boldsymbol{z})\right] \\
&= D_A(\boldsymbol{\phi}',\boldsymbol{\phi}).
\end{aligned}
$$

Plug the result back to Equation (8), we conclude that $Q_{\boldsymbol{\phi}}(\boldsymbol{\phi}')$ and the following are equivalent up to a constant independent of $\boldsymbol{\phi}'$:

$$
\frac{1}{|\mathcal{D}|}\sum_{\boldsymbol{x}\in\mathcal{D}}\log p_{\boldsymbol{\phi}}(\boldsymbol{x}) + \left\langle \frac{\partial \log p_{\boldsymbol{\phi}}(\boldsymbol{x})}{\partial \boldsymbol{\phi}},\boldsymbol{\phi}'-\boldsymbol{\phi}\right\rangle - \mathrm{KL}_{\boldsymbol{\phi}}(\boldsymbol{\phi}'). \tag{9}
$$

$\square$

We proceed to prove Lemma 1, which offers a practical way to compute the two key quantities in Equation (9).

*Proof of Lemma 1.* Recall from our definition that $p_{\boldsymbol{\phi}}(\boldsymbol{x}) := \tilde{p}_{\boldsymbol{\phi}}(\boldsymbol{x})/Z(\boldsymbol{\phi})$. We start by proving a key result: for each parameter $\phi_{n,c}$, the partition function

$$
Z(\boldsymbol{\phi}) = \mathrm{TD}(n)\cdot\exp(\phi_{n,c}) + C, \tag{10}
$$

where $C$ is independent of $\phi_{n,c}$. Note that by definition $Z(\boldsymbol{\phi})$ is computed by the same feedforward pass shown in Equation (1), with the only difference that the partition function is set to 1 for input nodes. Specifically, denote $Z_n(\boldsymbol{\phi})$ as the partition function of node $n$, we have

$$
Z_n(\boldsymbol{\phi}) = \begin{cases} 1 & n \text{ is an input node,} \\ \prod_{c\in\mathsf{ch}(n)} Z_c(\boldsymbol{\phi}) & n \text{ is a product node,} \\ \sum_{c\in\mathsf{ch}(n)}\exp(\phi_{n,c})\cdot Z_c(\boldsymbol{\phi}) & n \text{ is a sum node.} \end{cases}
$$

Define $\mathrm{TD}_m(n)$ as the TD-prob of node $n$ for the PC rooted at $m$ (assume $n$ is a descendant node of $m$). We prove Equation (10) by induction over $m$ in $Z_m(\boldsymbol{\phi})$.

In the base case where $m = n$, we have that

$$
\begin{aligned}
Z_m(\boldsymbol{\phi}) = Z_n(\boldsymbol{\phi}) &= \sum_{c'\in\mathsf{ch}(n)}\exp(\phi_{n,c'})\cdot Z_{c'}(\boldsymbol{\phi}), \\
&= \sum_{c'\in\mathsf{ch}(n)}\exp(\phi_{n,c'}), && \triangleright \text{ Since we assume } \forall c, Z_c(\boldsymbol{\phi}) = 1 \\
&= \exp(\phi_{n,c}) + \sum_{c'\in\mathsf{ch}(n),c'\neq c}\exp(\phi_{n,c'}), \\
&= \mathrm{TD}_m(n)\cdot\exp(\phi_{n,c}) + \sum_{c'\in\mathsf{ch}(n),c'\neq c}\exp(\phi_{n,c'}), && \triangleright \text{ Since } \forall c, \mathrm{TD}_c(c) = 1 \\
&= \mathrm{TD}_m(n)\cdot\exp(\phi_{n,c}) + C.
\end{aligned}
$$

Next, assume $m$ is a sum node and Equation (10) holds for all its children. That is,

$$
\forall b \in \mathsf{ch}(m), \quad Z_b(\boldsymbol{\phi}) = \mathrm{TD}_b(n)\cdot\exp(\phi_{n,c}) + C.
$$

We proceed by plugging in the definition of $Z_m(\boldsymbol{\phi})$:

$$
\begin{aligned}
Z_m(\boldsymbol{\phi}) &= \sum_{b\in\mathsf{ch}(m)}\exp(\phi_{m,b})\cdot Z_b(\boldsymbol{\phi}), \\
&= \sum_{b\in\mathsf{ch}(m)}\exp(\phi_{m,b})\cdot\mathrm{TD}_b(n)\cdot\exp(\phi_{n,c}) + C. \tag{11}
\end{aligned}
$$

Denote $\mathcal{A} \subseteq \mathsf{ch}(m)$ as the set of child nodes that are ancestors of $n$, and $\mathcal{B} = \mathsf{ch}(m)\backslash\mathcal{A}$ is its complement. From the definition of TD-probs, we have

$$
\begin{aligned}
\mathsf{TD}_m(n) &= \sum_{b\in\mathcal{A}} \mathsf{TD}_m(b) \cdot \mathsf{TD}_b(n), \\
&= \sum_{b\in\mathcal{A}} \exp(\phi_{m,b}) \cdot \mathsf{TD}_b(n). \qquad\qquad\qquad \triangleright \text{ Since } \mathsf{TD}_m(b) = \exp(\phi_{m,b})
\end{aligned}
$$

Plug in Equation (11), we conclude that

$$
\begin{aligned}
Z_m(\boldsymbol{\phi}) &= \sum_{b\in\mathcal{A}} \exp(\phi_{m,b}) \cdot \mathsf{TD}_b(n) \cdot \exp(\phi_{n,c}) + \sum_{b\in\mathcal{B}} \exp(\phi_{m,b}) \cdot \mathsf{TD}_b(n) \cdot \exp(\phi_{n,c}) + C, \\
&= \mathsf{TD}_m(n) \cdot \exp(\phi_{n,c}) + \sum_{b\in\mathcal{B}} \exp(\phi_{m,b}) \cdot \mathsf{TD}_b(n) \cdot \exp(\phi_{n,c}) + C, \\
&= \mathsf{TD}_m(n) \cdot \exp(\phi_{n,c}) + C'.
\end{aligned}
$$

Finally, if $m$ is a product node such that Equation (10) holds for all its children, we have that

$$
Z_m(\boldsymbol{\phi}) = \prod_{b\in\mathsf{ch}(m)} Z_b(\boldsymbol{\phi}).
$$

Since $m$ is decomposable (cf. Def. 3), there is at most one $b \in \mathsf{ch}(m)$ that is an ancestor of $n$ (otherwise multiple child nodes contain the variable scope of $n$). Denote that child node as $\hat{b}$, we further simplify the above equation to

$$
Z_m(\boldsymbol{\phi}) = Z_{\hat{b}}(\boldsymbol{\phi}) = \mathsf{TD}_{\hat{b}}(n) \cdot \exp(\phi_{n,c}) + C \tag{12}
$$

since all other terms are independent of $\phi_{n,c}$ and are assumed to be 1. According to the definition of TD-probs, we have

$$
\forall b \in \mathsf{ch}(m), \quad \mathsf{TD}_b(n) = \mathsf{TD}_m(n). \tag{13}
$$

Plug this into Equation (12) gives the desired result:

$$
Z_m(\boldsymbol{\phi}) = \mathsf{TD}_m(n) \cdot \exp(\phi_{n,c}) + C.
$$

This completes the proof of Equation (7).

We continue on proving the first equality in Lemma 1:

$$
\begin{aligned}
\frac{\partial \log p_{\boldsymbol{\phi}}(\boldsymbol{x})}{\partial \boldsymbol{\phi}} &= \frac{\partial \log \hat{p}_{\boldsymbol{\phi}}(\boldsymbol{x})}{\partial \boldsymbol{\phi}} - \frac{\partial \log Z(\boldsymbol{\phi})}{\partial \boldsymbol{\phi}}, \\
&= \frac{\partial \log \hat{p}_{\boldsymbol{\phi}}(\boldsymbol{x})}{\partial \boldsymbol{\phi}} - \frac{1}{Z(\boldsymbol{\phi})} \cdot \frac{\partial Z(\boldsymbol{\phi})}{\partial \boldsymbol{\phi}}, \\
&= \frac{\partial \log \hat{p}_{\boldsymbol{\phi}}(\boldsymbol{x})}{\partial \boldsymbol{\phi}} - \frac{\partial Z(\boldsymbol{\phi})}{\partial \boldsymbol{\phi}}.
\end{aligned}
$$

According to Equation (10), we can simplify the derivative of $Z(\boldsymbol{\phi})$ with respect to $\phi_{n,c}$ as $\mathsf{TD}(n) \cdot \exp(\phi_{n,c}) = \mathsf{TD}(\phi_{n,c})$, where the last equality follows from Definition 2. Therefore, we conclude that

$$
\frac{\partial \log p_{\boldsymbol{\phi}}(\boldsymbol{x})}{\partial \boldsymbol{\phi}} = \frac{\partial \log \hat{p}_{\boldsymbol{\phi}}(\boldsymbol{x})}{\partial \boldsymbol{\phi}} - \mathsf{TD}(\boldsymbol{\phi}).
$$

We move on to the second equality in Lemma 1. According to Vergari et al. [2021], $\mathsf{KL}_{\boldsymbol{\phi}}(\boldsymbol{\phi}')$ can be computed recursively as follows (define $\mathsf{KL}_{\boldsymbol{\phi}}^n(\boldsymbol{\phi}')$ as the KLD w.r.t. $n$):

$$
\mathsf{KL}_{\boldsymbol{\phi}}^n(\boldsymbol{\phi}') = \begin{cases} 0 & n \text{ is an input node,} \\ \sum_{c\in\mathsf{ch}(n)} \mathsf{KL}_{\boldsymbol{\phi}}^c(\boldsymbol{\phi}') & n \text{ is a product node,} \\ \sum_{c\in\mathsf{ch}(n)} \exp(\phi_{n,c})\big(\phi_{n,c} - \phi'_{n,c}\big) + \exp(\phi_{n,c}) \cdot \mathsf{KL}_{\boldsymbol{\phi}}^c(\boldsymbol{\phi}') & n \text{ is a sum node.} \end{cases} \tag{14}
$$

We want to show that for each $m$ that is an ancestor of $n$, the following holds:

$$\text{KL}_{\phi}^{m}(\phi') = -\text{TD}_m(n) \cdot \exp(\phi_{n,c}) \cdot \phi'_{n,c} + C, \tag{15}$$

where $C$ is independent of $\phi'_{n,c}$. We can use the exact same induction procedure that is used to prove Equation (10). For all ancestor sum nodes $m$ of $n$, the first term in Equation (14) (the last row among the three cases) is always independent of $\phi'_{n,c}$, and hence the recursive definition resembles that of $Z_m(\phi)$. Specifically, for all ancestor nodes of $n$, Equation (15) simplifies to

$$\text{KL}_{\phi}^{n}(\phi') = \begin{cases} 0 & n \text{ is an input node,} \\ \sum_{c \in \text{ch}(n)} \text{KL}_{\phi}^{c}(\phi') & n \text{ is a product node,} \\ \sum_{c \in \text{ch}(n)} \exp(\phi_{n,c}) \cdot \text{KL}_{\phi}^{c}(\phi') & n \text{ is a sum node.} \end{cases}$$

The key difference with $Z_n(\phi)$ is the definition of product nodes. Therefore, following the same induction proof of Equation (10), we only need to re-derive the case where $m$ is a product node such that Equation (15) holds for all its children.

Since the PC is decomposable, there is only one child node $b \in \text{ch}(m)$ that is an ancestor of $n$. Therefore, $\forall c \in \text{ch}(m), c \neq b$, $\text{KL}_{\phi}^{c}(\phi')$ is independent of $\phi'(n,c)$. Hence, we have

$$\begin{aligned} \text{KL}_{\phi}^{m}(\phi') &= -\text{TD}_b(n) \cdot \exp(\phi_{n,c}) \cdot \phi'_{n,c} + C, \\ &= -\text{TD}_m(n) \cdot \exp(\phi_{n,c}) \cdot \phi'_{n,c} + C. \qquad \triangleright \text{ According to Eq. (13)} \end{aligned}$$

Writing Equation (15) in a vectorized form for every $\phi'_{n,c}$ leads to our final result:

$$\text{KL}_{\phi}(\phi') = -\langle \text{TD}(\phi), \phi' \rangle + C.$$

$\square$

## B.2 DERIVATION OF THE FULL-BATCH AND MINI-BATCH EM

**Full-Batch EM.** Define $\mathcal{S}$ as the set of all sum nodes in the PC, the constrained optimization problem is

$$\begin{aligned} \underset{\phi'}{\text{maximize}} \quad & \left\langle \frac{1}{|\mathcal{D}|} \sum_{\boldsymbol{x} \in \mathcal{D}} \frac{\partial \log \tilde{p}_{\phi}(\boldsymbol{x})}{\partial \phi}, \phi' \right\rangle, \\ \text{s.t.} \quad & \forall n \in \mathcal{S}, \sum_{c \in \text{ch}(n)} \exp(\phi'_{n,c}) = 1. \end{aligned}$$

To incorporate the constraints, we use the method of Lagrange multipliers. The Lagrangian for this problem is

$$\mathcal{L}(\phi', \{\lambda_n\}_{n \in \mathcal{S}}) = \left\langle \frac{1}{|\mathcal{D}|} \sum_{\boldsymbol{x} \in \mathcal{D}} \frac{\partial \log \tilde{p}_{\phi}(\boldsymbol{x})}{\partial \phi}, \phi' \right\rangle + \sum_{n \in \mathcal{S}} \lambda_n \cdot \left( 1 - \sum_{c \in \text{ch}(n)} \exp(\phi'_{n,c}) \right),$$

where the Lagrange multipliers $\{\lambda_n\}_{n \in \mathcal{S}}$ enforce the constraints.

To minimize the Lagrangian w.r.t. $\phi'$, we take the partial derivative of $\mathcal{L}(\phi', \{\lambda_n\}_{n \in \mathcal{S}})$ w.r.t. each $\phi'_{n,c}$ and set it to 0:

$$\frac{\partial \mathcal{L}(\phi', \{\lambda_n\}_{n \in \mathcal{S}})}{\partial \phi'_{n,c}} = \text{F}_{\phi}^{\mathcal{D}}(n, c) - \lambda_n \exp(\phi'_{n,c}) = 0,$$

where $\text{F}_{\phi}^{\mathcal{D}}(n, c)$ is defined in Section 3.1. Simplifying this equation gives:

$$\phi'_{n,c} = \log \text{F}_{\phi}^{\mathcal{D}}(n, c) - \log Z,$$

where $Z = \sum_{c' \in \text{ch}(n)} \text{F}_{\phi}^{\mathcal{D}}(n, c')$.

**Mini-Batch EM.** Similar to the full-batch case, according to Section 3.2, the constrained optimization problem is

$$
\underset{\boldsymbol{\phi}'}{\text{maximize}} \left\langle \frac{1}{|\mathcal{D}|} \sum_{\boldsymbol{x} \in \mathcal{D}} \frac{\partial \log \tilde{p}_{\boldsymbol{\phi}}(\boldsymbol{x})}{\partial \boldsymbol{\phi}} + (\gamma - 1) \cdot \text{TD}(\boldsymbol{\phi}), \boldsymbol{\phi}' \right\rangle,
$$

$$
\text{s.t. } \forall n \in \mathcal{S}, \sum_{c \in \text{ch}(n)} \exp(\phi'_{n,c}) = 1.
$$

Following the full-batch case, the Lagrangian is given by

$$
\mathcal{L}(\boldsymbol{\phi}', \{\lambda_n\}_{n \in \mathcal{S}}) = \left\langle \frac{1}{|\mathcal{D}|} \sum_{\boldsymbol{x} \in \mathcal{D}} \frac{\partial \log \tilde{p}_{\boldsymbol{\phi}}(\boldsymbol{x})}{\partial \boldsymbol{\phi}} + (\gamma - 1) \cdot \text{TD}(\boldsymbol{\phi}), \boldsymbol{\phi}' \right\rangle + \sum_{n \in \mathcal{S}} \lambda_n \cdot \left( 1 - \sum_{c \in \text{ch}(n)} \exp(\phi'_{n,c}) \right).
$$

To minimize the Lagrangian with respect to $\boldsymbol{\phi}'$, we compute the partial derivative of $\mathcal{L}(\boldsymbol{\phi}', \{\lambda_n\}_{n \in \mathcal{S}})$ w.r.t. each $\phi'_{n,c}$ and set it equal to zero:

$$
\frac{\partial \mathcal{L}(\boldsymbol{\phi}', \{\lambda_n\}_{n \in \mathcal{S}})}{\partial \phi'_{n,c}} = \text{F}^{\mathcal{D}}_{\boldsymbol{\phi}}(n, c) + (\gamma - 1) \cdot \text{TD}(\phi'_{n,c}) - \lambda_n \exp(\phi'_{n,c}) = 0.
$$

Using the definition $\text{TD}(\phi'_{n,c}) = \text{TD}_{\boldsymbol{\phi}}(n) \cdot \exp(\phi_{n,c})$, the solution is given by

$$
\phi'_{n,c} = \log \left( \text{TD}_{\boldsymbol{\phi}}(n) \cdot \exp(\phi_{n,c}) + \alpha \cdot \text{F}^{\mathcal{D}}_{\boldsymbol{\phi}}(n, c) \right) - \log Z,
$$

where $\alpha := 1/(\gamma - 1)$ and $Z = \sum_{c \in \text{ch}(n)} \text{TD}_{\boldsymbol{\phi}}(n) \cdot \exp(\phi_{n,c}) + \alpha \cdot \text{F}^{\mathcal{D}}_{\boldsymbol{\phi}}(n, c)$.

## B.3   DECOMPOSITION OF PARAMETER FLOWS

In this section, we show that $\sum_{c \in \text{ch}(n)} \hat{\text{F}}^{\boldsymbol{x}}_{\boldsymbol{\phi}}(n, c) = 1$, where $n$ is a sum node. We start from the definition of $\hat{\text{F}}^{\boldsymbol{x}}_{\boldsymbol{\phi}}(n, c)$:

$$
\begin{aligned}
\hat{\text{F}}^{\boldsymbol{x}}_{\boldsymbol{\phi}}(n, c) &= \frac{\partial \log \tilde{p}^n_{\boldsymbol{\phi}}(\boldsymbol{x})}{\partial \phi_{n,c}} = \frac{1}{\tilde{p}^n_{\boldsymbol{\phi}}(\boldsymbol{x})} \cdot \frac{\partial \tilde{p}^n_{\boldsymbol{\phi}}(\boldsymbol{x})}{\partial \phi_{n,c}}, \\
&= \frac{\theta_{n,c}}{\tilde{p}^n_{\boldsymbol{\phi}}(\boldsymbol{x})} \cdot \frac{\partial \tilde{p}^n_{\boldsymbol{\phi}}(\boldsymbol{x})}{\partial \theta_{n,c}}, &&\triangleright \text{ By definition } \theta_{n,c} = \exp(\phi_{n,c}) \\
&= \frac{\theta_{n,c} \cdot \tilde{p}^c_{\boldsymbol{\phi}}(\boldsymbol{x})}{\tilde{p}^n_{\boldsymbol{\phi}}(\boldsymbol{x})}.
\end{aligned}
$$

Now we have

$$
\sum_{c \in \text{ch}(n)} \hat{\text{F}}^{\boldsymbol{x}}_{\boldsymbol{\phi}}(n, c) = \sum_{c \in \text{ch}(n)} \frac{\theta_{n,c} \cdot \tilde{p}^c_{\boldsymbol{\phi}}(\boldsymbol{x})}{\tilde{p}^n_{\boldsymbol{\phi}}(\boldsymbol{x})} = 1.
$$

## C   GLOBAL PARAMETER RENORMALIZATION OF PCS

In this section, we propose a simple renormalization algorithm that takes an unnormalized PC $p_{\boldsymbol{\phi}}(\mathbf{X})$ (i.e., its partition function does not equal 1) with parameters $\boldsymbol{\phi}$ and returns a new set of parameters $\boldsymbol{\phi}'$ such that for each node $n$ in the PC

$$
\forall \boldsymbol{x}, \; p^n_{\boldsymbol{\phi}'}(\boldsymbol{x}) = \frac{1}{Z_n(\boldsymbol{\phi})} \cdot p^n_{\boldsymbol{\phi}}(\boldsymbol{x}),
$$

where $Z(\boldsymbol{\phi}) := \sum_{\boldsymbol{x}} p^n_{\boldsymbol{\phi}}(\boldsymbol{x})$ is the partition function of $p^n_{\boldsymbol{\phi}}$.

**The Algorithm.**  First, we perform a feedforward evaluation of the PC to compute the partition function $Z_n(\phi)$ of every node $n$:

$$
Z_n(\phi) = \begin{cases} 1 & n \text{ is an input node,} \\ \prod_{c \in \mathsf{ch}(n)} Z_c(\phi) & n \text{ is a product node,} \\ \sum_{c \in \mathsf{ch}(n)} \exp(\phi_{n,c}) \cdot Z_c(\phi) & n \text{ is a sum node.} \end{cases}
$$

Next, for every sum edge $(n, c)$ (i.e., $n$ is a sum node and $c$ is one of its children), we update the parameter as

$$
\phi'_{n,c} = \log \left( \frac{\theta_{n,c} \cdot Z_\phi(c)}{Z_\phi(n)} \right), \tag{16}
$$

where $\theta_{n,c} := \exp(\phi_{n,c})$. The existence of this normalization algorithm has been previously shown by Martens and Medabalimi [2014], although they did not provide an easy-to-implement algorithm.

**Analysis.**  We begin by proving the correctness of the algorithm. Specifically, we show by induction that $p_{\phi'}^n(x) = p_\phi^n(x)/Z_\phi(n)$ for every $n$ and $x$. In the base case, all input nodes satisfy the equation since they are assumed to be normalized. Next, given a product node $n$, assume the distributions encoded by all its children $c$ satisfy

$$
\forall c \in \mathsf{ch}(n), \ p_{\phi'}^c(x) = p_\phi^c(x)/Z_\phi(c). \tag{17}
$$

Then by definition, $p_{\phi'}^n(x)$ can be written as:

$$
\begin{aligned}
p_{\phi'}^n(x) &= \prod_{c \in \mathsf{ch}(n)} p_{\phi'}^c(x) = \prod_{c \in \mathsf{ch}(n)} p_\phi^c(x)/Z_\phi(c), \\
&= \frac{\prod_{c \in \mathsf{ch}(n)} p_\phi^c(x)}{\prod_{c \in \mathsf{ch}(n)} Z_\phi(c)}, \\
&= \frac{p_\phi^n(x)}{Z_\phi(n)}.
\end{aligned}
$$

Finally, consider a sum node $n$ whose children satisfy Equation (17). We simplify $p_{\phi'}^n(x)$ in the following:

$$
\begin{aligned}
p_{\phi'}^n(x) &= \sum_{c \in \mathsf{ch}(n)} \theta'_{n,c} \cdot p_{\phi'}^c(x), \\
&= \sum_{c \in \mathsf{ch}(n)} \frac{\theta_{n,c} \cdot Z_\phi(c)}{Z_\phi(n)} \cdot p_{\phi'}^c(x), && \triangleright \text{According to Eq. (16)} \\
&= \sum_{c \in \mathsf{ch}(n)} \frac{\theta_{n,c} \cdot \cancel{Z_\phi(c)}}{Z_\phi(n)} \cdot \frac{p_\phi^c(x)}{\cancel{Z_\phi(c)}}, && \triangleright \text{By induction} \\
&= \frac{\sum_{c \in \mathsf{ch}(n)} \theta_{n,c} \cdot p_\phi^c(x)}{Z_\phi(n)}, \\
&= p_\phi^n(x)/Z_\phi(n).
\end{aligned}
$$

We proceed to show an interesting property of the proposed global renormalization.

**Lemma 2.**  *Given a PC $p_\phi(\mathbf{X})$. Denote $\phi'$ as the parameters returned by the global renormalization algorithm. Then, for every sum edge $(n, c)$, we have*

$$
\forall x, \ \frac{\partial \log p_{\phi'}(x)}{\partial \phi'_{n,c}} = \frac{\partial \log p_\phi(x)}{\partial \phi_{n,c}}.
$$

*Proof.*  We begin by showing that for each node $n$ is one of its children, the following holds:

$$
\forall x, \ \frac{\partial \log p_{\phi'}^n(x)}{\partial \log p_{\phi'}^c(x)} = \frac{\partial \log p_\phi^n(x)}{\partial \log p_\phi^c(x)}.
$$

If $n$ is a product node, both the left-hand side and the right-hand side equal 1. For example, consider the left-hand side. According to the definition, we have

$$\log p_{\phi'}^n(\boldsymbol{x}) = \sum_{c \in \mathsf{ch}(n)} \log p_{\phi'}^c(\boldsymbol{x}).$$

Hence, its derivative w.r.t. $\log p_{\phi'}^c(\boldsymbol{x})$ is 1.

If $n$ is a sum node, then for each $\boldsymbol{x}$, we have

$$
\begin{aligned}
\frac{\partial \log p_{\phi'}^n(\boldsymbol{x})}{\partial \log p_{\phi'}^c(\boldsymbol{x})} &= \frac{p_{\phi'}^c(\boldsymbol{x})}{p_{\phi'}^n(\boldsymbol{x})} \cdot \frac{\partial p_{\phi'}^n(\boldsymbol{x})}{\partial p_{\phi'}^c(\boldsymbol{x})}, \\
&= \frac{p_{\phi'}^c(\boldsymbol{x})}{p_{\phi'}^n(\boldsymbol{x})} \cdot \theta'_{n,c}, \\
&= \frac{p_\phi^c(\boldsymbol{x})/Z_\phi(c)}{p_\phi^n(\boldsymbol{x})/Z_\phi(n)} \cdot \theta'_{n,c}, \\
&= \frac{p_\phi^c(\boldsymbol{x})/\cancel{Z_\phi(c)}}{p_\phi^n(\boldsymbol{x})/\cancel{Z_\phi(n)}} \cdot \frac{\theta_{n,c} \cdot \cancel{Z_\phi(c)}}{\cancel{Z_\phi(n)}}, \\
&= \frac{p_\phi^c(\boldsymbol{x})}{p_\phi^n(\boldsymbol{x})} \cdot \theta_{n,c}, \\
&= \frac{p_\phi^c(\boldsymbol{x})}{p_\phi^n(\boldsymbol{x})} \cdot \frac{\partial p_\phi^n(\boldsymbol{x})}{\partial p_\phi^c(\boldsymbol{x})}, \\
&= \frac{\partial \log p_\phi^c(\boldsymbol{x})}{\partial \log p_\phi^n(\boldsymbol{x})}.
\end{aligned}
$$

$\square$

## D EMPIRICAL EVALUATIONS

We train an HCLT [Liu and Van den Broeck, 2021] with hidden size 512 on $16 \times 16$ aligned patches from the ImageNet32 dataset [Deng et al., 2009]. That is, we partition every $32 \times 32$ image (there are three color channels) into four $16 \times 16$ patches and treat these as data samples. There are in total $16 \times 16 \times 3 = 768$ categorical variables in the PC.

We apply a lossy YCC transformation proposed by Malvar and Sullivan [2003]. Specifically, given a pixel with RGB values $(R, G, B)$, we first normalize them to the range $[0, 1]$ by

$$r = R/255, \ g = G/255, \ b = B/255.$$

We then apply the following linear transformation:

$$co = r - b, \ tmp = b + co/2, \ cg = g - tmp, \ y = tmp * 2 + cg + 1,$$

where $y$, $co$, and $cg$ are all in the range $[-1, 1]$. Finally, we quantize the interval $[-1, 1]$ into 256 bins uniformly and convert $y$, $co$, and $cg$ to their quantized version $Y$, $Co$, and $Cg$, respectively. Note that $Y$, $Co$, and $Cg$ are all categorical variables with 256 categories.

**Full-Batch EM.** The full-batch EM implementation follows prior work (e.g., Choi et al. [2021], Peharz et al. [2020]).

**Mini-Batch EM.** For notation simplicity, we rewrite Equation (6) as

$$\theta'_{n,c} = \left( (1 - \alpha) \cdot \mathrm{TD}_\phi(n) \cdot \theta_{n,c} + \alpha \cdot \mathrm{F}_\phi^{\mathcal{D}}(n, c) \right) / Z, \tag{18}$$

which makes it more consistent with the baseline mini-batch EM algorithm. For both our algorithm and the baseline algorithm in Equation (5), we choose a base learning rate of $\alpha = 0.4$ and a cosine learning rate decay schedule to decrease it

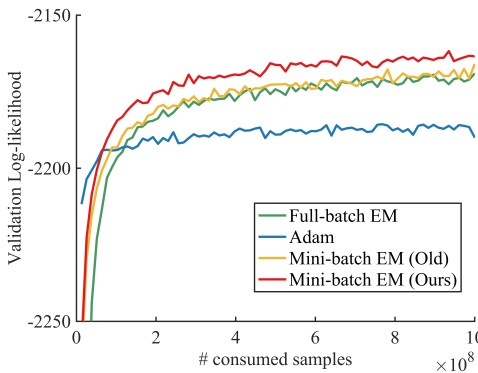

Figure 2: Validation log-likelihood of an HCLT PC [Liu and Van den Broeck, 2021] on $16 \times 16$ patches from ImageNet32. Our proposed mini-batch EM algorithms outperform all other optimizers by a large margin.

to $\alpha = 0.08$. For both algorithms, we use a batch size of $32768$. We also use momentum to update the flows $\texttt{F}_\phi^\mathcal{D}$. Specifically, we initialize the momentum flows $\texttt{Fm}_\phi^\mathcal{D} = \mathbf{0}$, then before every EM step, we update $\texttt{Fm}_\phi^\mathcal{D}$ following:

$$\texttt{Fm}_\phi^\mathcal{D} \leftarrow \eta \cdot \texttt{Fm}_\phi^\mathcal{D} + (1 - \eta) \cdot \texttt{F}_\phi^\mathcal{D}.$$

Finally, we replace the $\texttt{F}_\phi^\mathcal{D}(n, c)$ in Equation (18) with $\texttt{Fm}_\phi^\mathcal{D}(n, c)/(1 - \eta^{T+1})$, where $T$ is the number of updates performed.

**Gradient-Based.** Following Loconte et al. [2025, 2024], we adopt the Adam optimizer [Kingma, 2014]. We tested with learning rates $1e-2$, $3e-3$, and $1e-3$, and observed that $1e-2$ performed the best. This matches the observation in Loconte et al. [2024]. We also tested different batch sizes, and $1024$ performs the best given a fixed number of epochs.

**Empirical Insights.** To have a fair comparison between full-batch and mini-batch algorithms, we plot the validation log-likelihood vs. the total number of consumed samples. Results are shown in Figure 2. First, we observe that EM-based algorithms perform better than the gradient-based method Adam, suggesting that EM is a better parameter learning algorithm for PCs. Next, the proposed mini-batch EM algorithm outperforms both the full-batch EM algorithm and the mini-batch EM algorithm proposed by Peharz et al. [2020], indicating the empirical superiority of the proposed algorithm.