# OpenReview forum: "Rethinking Probabilistic Circuit Parameter Learning"
_auai.org/UAI/2025/Workshop/TPM — TPM 2025_

### Official Review · Reviewer_MLeu · 2025-06-08
**Investigating EM for PC parameter learning**

**Rating:** 3

**Review:**

This paper brings to light a critical gap in the parameter learning of Probabilistic Circuits (PCs), by providing the first principled derivation of a mini-batch Expectation-Maximization (EM) algorithm.
It uncovers the hidden connection between the classical EM algorithm for PCs and a principled, mini-batch-friendly update rule. The authors show that full-batch EM is really just maximizing a regularized, first-order log-likelihood approximation, and provide a recipe for scaling EM to large datasets without heuristic tweaks but just using a closed-form, easy-to-implement EM update.

---

### Official Review · Reviewer_EBSY · 2025-06-11

**Rating:** 3

**Review:**

The paper proposes a new mini-batch expectation maximization (EM) algorithm for PCs that is derived directly from the full-batch EM by reweighting the KL divergence term. Each step in the new mini-batch EM is just as efficient as similar mini-batch EM variants (Peharz, 2015) but incorporate the relative influence of each node in the output. Empirically, the new method outperforms other EM variants as well as Adam.

### Strengths

- I see optimization as one of the main open problems in PCs, and it is refreshing to see a paper tackling that directly.  Learning tends to converge quite quickly but seems to get stuck in poor local minima quite often.
- The new mini-batch EM algorithm is elegant, intuitive and seems to provide better empirical performance too. The term capturing the relative importance of each node (equation at the bottom left of page 4) is particularly interesting.
- The mathematical development seems sound and clear, although I admit not having checked all the proofs in the appendix.

### Weaknesses

- The empirical evaluation is limited to a similar experiment.
- Some related work is missing. Notably, one of the approaches to sidestep the difficulties in PC optimization has been to introduce continuous latent variables during optimization, replacing sum nodes with integral nodes (Correia et al. 2023, Gala et al. 2024). For future versions of the paper, it would be interesting to compare how this new version of mini-batch EM compares to these methods.

### Questions
1. It seems that the new mini-batch EM outperforms full-batch EM. Is that fully explained by the stochasticity in the mini-batching, as in SGD?

### Small remarks

It seems $\mathcal D$ is used for both the entire dataset and a mini-batch. I think it would be clearer if we had a different notation for each.

### References

Correia, Alvaro HC, et al. "Continuous mixtures of tractable probabilistic models." Proceedings of the AAAI Conference on Artificial Intelligence. Vol. 37. No. 6. 2023.

Gala, Gennaro, et al. "Probabilistic integral circuits." International Conference on Artificial Intelligence and Statistics. PMLR, 2024.

Robert Peharz. Foundations of sum-product networks for probabilistic modeling. PhD thesis, PhD thesis, Medical University of Graz, 2015.

**Nominate For Best Paper:**

["Yes"]